# Building Trust and Awareness to Increase AZ Native Nation Participation in COVID-19 Vaccines

**DOI:** 10.3390/ijerph20010031

**Published:** 2022-12-20

**Authors:** Grant Sears, Marissa Tutt, Samantha Sabo, Naomi Lee, Nicolette Teufel-Shone, Anthony Baca, Marianne Bennett, J. T. Neva Nashio, Fernando Flores, Julie Baldwin

**Affiliations:** 1Center for Health Equity Research, Northern Arizona University, Flagstaff, AZ 86011, USA; 2River People Health Center, Scottsdale, AZ 85256, USA; 3White Mountain Apache Tribe CHR Program, Whiteriver, AZ 85941, USA; 4Colorado River Indian Tribes, Parker, AZ 85344, USA

**Keywords:** American Indian, coronavirus, health education, community health representative, community health worker, vaccine hesitancy

## Abstract

The goal of this study was to establish effective, culturally appropriate strategies to enhance participation of American Indian/Alaska Native (AI/AN) communities in prevention and treatment of COVID-19, including vaccine uptake. Thirteen Community Health Representatives (CHRs) from three Arizona Native nations tailored education materials to each community. CHRs delivered the intervention to over 160 community members and administered a pre-posttest to assess trusted sources of information, knowledge, and self-efficacy and intention regarding COVID-19 vaccines. Based on pre-posttest results, doctors/healthcare providers and CHRs were the most trusted health messengers for COVID-19 information; contacts on social media, the state and federal governments, and mainstream news were among the least trusted. Almost two-thirds of respondents felt the education session was relevant to their community and culture, and more than half reported using the education materials to talk to a family member or friend about getting vaccinated. About 67% trusted the COVID-19 information provided and 74% trusted the CHR providing the information. Culturally and locally relevant COVID-19 vaccine information was welcomed and used by community members to advocate for vaccination. The materials and education provided by CHRs were viewed as helpful and emphasized the trust and influence CHRs have in their communities.

## 1. Introduction

The SARS-CoV-2 pandemic, henceforth referred to as COVID-19, presented challenges in combating transmission, morbidity, and mortality at local and national levels. These challenges are amplified for American Indian/Alaska Native (AI/AN) populations, who face one of the highest rates of COVID-19-related discrimination [1]. Lasting effects from historical trauma, persistent inequity in healthcare access, and difficulty in obtaining other basic necessities, such as running water, make coping with the pandemic a challenge. In the first six months of the pandemic, COVID-19 incidence among AI/AN populations was 3.5 times higher than for white persons [2,3]. Vaccines remain one of the safest and most convenient tools of modern medicine in mitigating loss of life and illness during these outbreaks. Vaccinations are safe and convenient prevention for both children and adults; they can lower the chance of getting a disease and spreading it by producing immunity but not disease. Typically, vaccines require years of testing and monitoring prior to the Food and Drug Administration authorizing them for use [4]. Given the widespread prevalence and severity of COVID-19, as well as advancements in medicine and technology, vaccines were developed and employed much faster for this pandemic. It was reasonable to expect hesitancy regarding the COVID-19 vaccine given their quick development. Additionally, rapid sharing of misinformation via television, websites, and social medias, coupled with confirmation bias, can exacerbate vaccine hesitancy [5,6]. It is important to give credible health information and be transparent when communicating about the COVID-19 vaccines, and especially to AI/AN communities who were not sufficiently represented in vaccine trials [7] and remain cautious towards research [8].

American Indians in Arizona have taken extraordinary actions to protect their people and to prevent the worst outcome of the COVID-19 pandemic. Many Native nations implemented mask mandates, curfews and stay-at-home orders, and closed access to tribal lands/resources in efforts to mitigate the impact of COVID-19 in their communities [9]. Organized AI/AN efforts to combat COVID-19 have shown community-based values enacted through resource sharing and innovative solutions, such as telephonic check-ins and zero-contact aid pickup [10]. Rates of COVID-19 infection vary significantly among Arizona’s AI/AN; some Native nations reported few cases while others experienced some of the highest infection rates per capita in the U.S. [11].

In July 2020, the National Institutes of Health (NIH) launched the COVID-19 Prevention Trials Network (CoVPN) by merging four existing NIAID-funded clinical trials networks [12]. To increase community engagement, especially for those communities of color most at-risk, the CoVPN project coordinators assembled four panels of experts to review each of the Phase III COVID-19 vaccine study protocols, including education materials, informed consent forms, and early Phase I and II study data. Key themes identified by the panel included concerns regarding data sovereignty when working with Native nations, appropriate methods to engage AI/AN communities and tribal leaders, limited locations and access for AI/AN members in rural communities, and the lack of culturally tailored education tools and modalities of dissemination. Furthermore, a national assessment of COVID-19 vaccine hesitancy in the American adult population concluded evidence-based communication and media strategies were crucial to reach mass immunization [13]. Rural dwellers, those with lower income, and those with a lower perceived threat of infection may require special attention, as they exhibited greater vaccine hesitancy than their urban dwelling or higher income counterparts [13].

Given the dearth of culturally tailored COVID-19 education tools for AI/AN communities, the goal of this project was to establish effective, culturally appropriate strategies to enhance participation of AI/AN communities in the prevention and treatment of COVID-19, including vaccine uptake and education.

## 2. Materials and Methods

### 2.1. Tribal Partners

Three Arizona Native nations with a long-standing relationship with members of the research team at Northern Arizona University (NAU)’s Center for Health Equity Research (CHER) were invited to participate in the project. Native nations included Salt River Pima-Maricopa Indian Community, White Mountain Apache Tribe, and Colorado River Indian Tribes; each operated CHR programs active in addressing COVID-19. A Community Health Representative can act as a liaison for community members in achieving their healthcare needs. They have become essential in tribal community-oriented healthcare services [14]. The results presented below are representative of only the respondents in these three communities, they may not generalize to other populations. Of the 22 federally recognized Native nations in Arizona, however, 19 operate a CHR program [15].

The Salt River Pima-Maricopa Indian Community (SRPMIC), located in southern Arizona in the metropolitan Phoenix area, covers 52,600 acres, with 19,000 held as natural preserve. The community operates a full-service government and oversees departments, programs, projects, and facilities. SRPMIC has over 10,800 enrolled members. The White Mountain Apache Tribe (WMAT), located in eastern Arizona, covers 1.6 million acres. There are approximately 15,000 enrolled tribal members, many of whom live on their Tribal lands but also throughout the country and overseas. The Colorado River Indian Tribes (CRIT), located in western Arizona along the Arizona/California border, is made up of 4 tribes: the Mohave, Chemehuevi, Hopi, and Navajo. The reservation was created for the Mohave and Chemehuevi since they inhabited the area for centuries, then people from Hopi and Navajo were relocated to the reservation in later years. There are about 4300 enrolled tribal members and CRIT spans nearly 300,000 acres of land.

### 2.2. Project Goals and Objectives

This project was funded from the National Institute of Minority Health and Health Disparities (NIMHD), grant no. 3U54MD012388-04S5. Aims of this project included the following: Aim (1) Assess awareness, knowledge, experiences, concerns, attitudes, and needs regarding COVID-19 vaccine uptake among AI/AN communities in Arizona; Aim (2) Develop and adapt culturally-appropriate education materials and strategies designed to increase awareness of COVID-19 vaccines, decrease misinformation, and increase medical trust; Aim (3) Implement the education session and evaluate the impact of education materials and strategies on enhanced awareness, trust, self-efficacy, and willingness. Knowledge regarding COVID-19 vaccines, trusted health messengers, and self-efficacy and intention to vaccinate were outcome variables assessed through pre-postsurvey methodology. Acceptability, feasibility, and participant satisfaction with the brief intervention were assessed as well.

This project was grounded in community-based participatory research [16] and used a consensus panel and survey methodology. Consensus panels are a highly participatory approach to gaining rapid feedback; an approach study personnel deemed fitting for feedback on COVID-19 health education materials developed for community members. Thirteen (13) CHRs were recruited across the three tribes in Arizona to both participate in the consensus panels and conduct this brief intervention. CHRs were trained to provide a brief education session with accompanying handouts to members of their respective communities regarding COVID-19 health, safety, and vaccine information. A presurvey to assess attitudes, knowledge, and self-efficacy regarding COVID-19 vaccines was completed by the participant at baseline. A follow-up postsurvey was completed 1 month later.

### 2.3. Recruitment and Incentives

The three tribes were approached to participate in this study based on opportunities to build upon existing partnerships and the geographic diversity between the three implementation sites. To honor the Tribal Consultation Policy of the Arizona Board of Regents, pre-established institutional guidelines for working with Native nations in research projects were followed [17]. Leaders from the three participating tribes either signed a Memorandum of Agreement, provided a Tribal Resolution of Support, or a letter from Tribal Council to participate in this project. CHR managers and their CHRs at each tribe were presented project details. Participating CHRs were consented and enrolled into the project at the time of the first consensus panel. All CHRs and community participants were (a) age 18 years or older and less than 100 years; (b) an enrolled member of one of these three Arizona Native nations; (c) without significant cognitive impairment; (d) capable of providing informed consent and participating in study activities; and (e) English speakers. Additionally, CHR participants needed to self-identify as a member of the CHR workforce, which are known by many titles, like Promotora, Community Health Worker, Community Health Advisor or Patient Navigator. *Exclusion criteria*: (a) younger than 18 years of age; (b) current participant in another intervention-based COVID-19 vaccine research study; (c) prisoner; or (d) with cognitive impairment. The goal was to recruit 1800 community participants between the three Native nations and their CHRs; the reality was significantly fewer (<200).

Although CHRs were not compensated financially for their involvement in the project, they were each given an 8th generation Apple iPad tablet to use for project activities and to retain after data collection was finished. The CHR programs were also compensated USD 500 plus USD 100 per participating CHR. Community survey participants were provided USD 10 gift cards to a local grocery store for each pre- and posttest survey they completed.

### 2.4. Consensus Panels and Education Materials Development

Consensus panel methodology was used for the tailoring of education materials. Two consensus panels were conducted for each tribe; they were scheduled two weeks apart, conducted via Zoom^™^ telecommunications platform, and lasted no longer than two hours each session. Prior to the first consensus panel, NAU research coordinators drafted education materials based on information and guidelines available via the Centers for Disease Control and Prevention, Johns Hopkins Center for American Indian Health, and the Arizona Department of Health Services. The purpose of the first consensus panel was to present CHRs with education materials and get their thoughts and recommendations based on the handouts’ content (i.e., language and information), format/organization, and imagery (i.e., colors and graphics). During the consensus panel, one research coordinator guided the conversation while a second presented the education material on screen and took note of CHR comments and recommendations. Following the first consensus panel, information was tailored to best address the concerns or needs of the community identified by the CHRs. Research coordinators made revisions based on the feedback from consensus panel one. During consensus panel two researchers presented the revised handouts to CHRs and facilitated a discussion to make final edits and seek validation before printing. A total of four handouts were developed for this project: (1) COVID-19 vaccine information, (2) COVID-19 frequently asked questions and myths, (3) clinical trial information, and (4) grief and coping (note: grief and coping was an additional handout developed for one tribe, per their request). See Appendix A. Handouts were updated and reprinted over time in response to evolving tribal, state, and federal guidelines regarding COVID-19.

### 2.5. Survey Development and REDCap^®^ Integration

This project utilized a baseline (pretest) and follow-up (posttest) survey methodology; the posttest was administered 30 days after the baseline. The surveys included questions that addressed the following project outcome variables: (a) acceptability: at follow-up, participants rated their acceptability of the education session and the cultural relevancy of the materials on a 5-point Likert scale; (b) feasibility: the feasibility of the intervention was assessed in relation to the ability to recruit and survey 1800 participants; (c) participant satisfaction ratings: at follow-up, participants rated global satisfaction of the intervention they received on a 5-point Likert scale; (d) level of trust in different sources of information to provide accurate COVID-19 information; (e) knowledge about COVID-19 vaccines and vaccine trials; and (f) self-efficacy and intention to participate in COVID-19 vaccine trials and get vaccinated. All outcomes, other than feasibility, were assessed through an adapted version of the CEAL Revised Common Survey, sourced from the NIH PhenX Toolkit, a collection of COVID-19 related measurement protocols [12].

The surveys were first developed on paper and then made in digital format in REDCap^®^ (v. 12.4.18), Research Electronic Data Capture, a secure web platform for building and managing surveys and datasets. User accounts were created for each CHR and tethered to the project server. Each CHR was assigned to a “Data Access Group” and given limited user rights to upload data. Using the REDCap Mobile App downloaded to 8th generation Apple^®^ iPad (v. iPadOS 15.7.2) tablets, CHRs could survey community members and upload data to the project server. REDCap user accounts coupled with tablets assigned to each CHR streamlined data collection and analysis. After data collection had concluded, research coordinators administered a brief questionnaire to CHRs. The purpose of the questionnaire was to understand, from their perspective, their feelings about the process of the project and what could have been improved. CHRs were asked five questions regarding strategies for recruitment, strategies for follow-up, why individuals did not complete the post-survey, how the project could be improved, and any other comments.

### 2.6. CHR Training and Support

After the project server, user accounts, and surveys were created, and education materials printed, project personnel traveled to each site to conduct a 2 h in-person training with CHRs. Precautions were taken to prevent COVID-19 transmission. These face-to-face sessions were excellent opportunities for research coordinators to meet and build connections with the CHRs. iPads were assigned to CHRs and their user accounts synced to the REDCap project server at this time. Research coordinators explained the handouts that CHRs received and demonstrated data collection procedures (i.e., surveying, providing incentive, uploading data, and maintaining logs). CHRs were encouraged to present the education material as they saw fit, as they are recognized and trusted assets within their communities. Project personnel maintained an open line of communication for questions and problem solving throughout implementation. Questions regarding data collection and uploading procedures using the tablets were common.

### 2.7. Data Collection and Analysis

Using the iPad assigned to them, CHRs administered the pretest at baseline, prior to providing the COVID-19 handouts and education session. A follow-up posttest was administered either in-person or telephonically approximately 1 month after baseline. CHRs were encouraged to start with their existing client lists for their outreach. As client lists were exhausted, CHRs were encouraged to use high-traffic areas in their community such as local grocers or food vendors where they could recruit participants for the brief survey and education session. REDCap Mobile App enabled CHRs to collect survey data offline without the need for an internet connection; an internet connection was required to upload survey data to the project server. CHRs were encouraged to upload their data at the end of the day or week when they returned to their office and had a reliable internet connection. Throughout the 8-month implementation period (April–November 2021), data were stored on NAU’s secure REDCap project server. After data collection was finalized, the database was exported to IBM SPSS Statistical Software^®^ (v. 28.0; SPSS). Data were cleaned and analyzed in SPSS software downloaded on secure NAU computers. Additionally, a brief exit questionnaire was administered to CHRs to gather their feedback on the project and successful recruitment strategies. Results are summarized below.

## 3. Results

### 3.1. Demographics

The mean age of respondents was 46 years old. Sixty percent (60%) of participants identified as women, 37% men, and the remaining identified as nonbinary, genderqueer, genderfluid, or they preferred not to answer. Ninety-two percent (92%) identified as American Indian or Alaska Native, 4% identified as white, 3% preferred not to answer, 2% other, and less than 1% identified as black or African American. Ten percent (10%) of respondents identified as Hispanic or Latino, leaving 83% and 7% as non-Hispanic or Latino and prefer not to answer, respectively. At baseline, 24% of households had no members vaccinated against COVID-19 and nearly half (47%) had at least one member get a positive COVID-19 diagnosis at some point in the pandemic. See Table 1 below for more detail.

### 3.2. Trusted COVID-19 Health Messengers

The most trusted health messengers among respondents included their doctor/healthcare provider and their CHR. More than a third trusted both their healthcare provider and CHR “a great deal” and an additional 24% trusted them “a little,” as opposed to feeling neutral and/or “not at all” trusting. Least trusted health messengers included contacts on social media, the U.S. government, mainstream news, and the Arizona state government. Nearly half (49%) did not trust their contacts on social media at all, and about one-third did not trust the federal and Arizona state governments, 35% and 32%, respectively. Only about 18% trusted the Arizona state government regarding COVID-19 information and thirteen percent (13%) trusted the federal government. Close friends and family members, workmates/classmates, and faith leaders were all sources of COVID-19 information in which most respondents felt “neutral”, but leaning more towards trusting than not. One-third (34%) of respondents did not trust news on the radio, TV, online, or in newspapers at all; however, most felt neutral. Although local and neighboring tribal governments were trusted more than the state and federal, most respondents remained neutral towards these sources. See Table 2 below for more detail.

### 3.3. Vaccine Confidence

At baseline, two-thirds (66%) of participants were vaccinated against COVID-19. The overwhelming majority (96%) of those vaccinated had received both doses of the Moderna or Pfizer vaccine. This trend continued through the 30-day follow-up survey. The top three reasons in support of getting a COVID-19 vaccine reported at baseline included “I want to keep my family safe” (74%), “I want to keep myself safe” (61%), and “I want to keep my community safe” (51%). At follow-up, these remained the most reported reasons why someone would get a COVID-19 vaccine. The top three reasons against getting a vaccine reported at baseline included “I’m concerned about the side effects from the vaccine” (39%), “I don’t know enough about how well the COVID-19 vaccine works” (35%), and “I don’t trust that the vaccine will be safe” (21%). Although concerns about vaccine safety and side-effects remained the top reported reasons against vaccination at follow-up, they were reported in lower frequencies. The most common reason for why a respondent had not been vaccinated between baseline and follow-up was because they had been fully vaccinated against COVID-19 before the education session (14%). Knowledge regarding COVID-19 vaccines generally improved among all participants. Specifically, participants increased their understanding of and their ability to explain how the vaccine worked, the side-effects, and how to arrange for a vaccine for themselves or a family member. Knowledge/Ability questions were answered on a 5-point Likert scale ranging from (5) High, (4) Somewhat high, (3) Neutral, (2) Somewhat low, to (1) Low. See Table 3 and Table 4 below for more detail.

### 3.4. Acceptability and Satisfaction with Intervention

More than half (56%) of respondents reported using the COVID-19 education handouts to talk to a family member or friend about getting vaccinated since they had received them. Overall, 72% of respondents felt satisfied with the COVID-19 education session. About 64% agreed the education session felt relevant to their community and culture; 67% agreed the information provided was trustworthy. Nearly three-quarters of respondents (74%) trusted the educator providing the information, that being the CHR. Seventy-seven percent (77%) agreed the information provided was easy to understand and 60% agreed that the education session gave new information about COVID-19 that was not previously known. See Table 5 below for more detail.

### 3.5. CHR Questionnaire and Project Reflection

CHRs reported that the following strategies enhanced recruitment: (1) beginning recruitment through existing client lists, (2) as the client lists were exhausted, recruiting through family and friends, (3) going house-to-house in the community, and (4) setting up at high traffic areas in their community. The following strategies led to better follow-up: (1) CHRs saw a lot of their participants on a day-to-day basis, which helped in reminding them about their follow up survey and scheduling a follow-up appointment with their participant during the first session and (2) calling to remind them when their 1-month follow-up survey was approaching. Although COVID-19 vaccination rates were high at the beginning of the project, CHRs reported that they were able to support hesitant people to get vaccinated using the materials developed through this program.

## 4. Discussion

The results from this study demonstrate the positive impact that effective, culturally tailored health education materials can have within AI/AN communities coping with a pandemic, and especially so when delivered from a CHR. The COVID-19 education handouts increased respondents’ understanding of and their ability to explain how the COVID-19 vaccine works, the side-effects of the vaccine, and how to schedule vaccination for themselves or a family member. Primary care providers and CHRs were among the most trusted sources of COVID-19 health information. Half of respondents did not trust social media contacts at all; about one-third did not trust the U.S. federal government, mainstream news, or Arizona state government. Respondents felt more trusting of local and neighboring tribal governments, as opposed to state and federal, but felt neutral about them overall. Reasons that motivated respondents to get a COVID-19 vaccine were focused on keeping themselves, their family, and their community safe. Reasons that deterred individuals from vaccination were not knowing how the vaccine works, concern about side-effects, and a lack of trust in the safety of the vaccine. Reasons against vaccination were reported in significantly lower frequencies at follow-up than at baseline. This may be explained by the education materials provided and the subsequent increase in participants’ knowledge regarding how the vaccine functions, its side-effects, and how to schedule a vaccination locally. A waning sense of urgency/interest may also explain this phenomenon. More than half of respondents used the provided Handouts to talk to a family member or friend about getting vaccinated; they favorably received the COVID-19 education materials and especially liked the CHRs providing the information. Considering the trust and impact CHRs have in their communities, and that the majority of Arizona Native nations employ CHR programs [15], they are recognized as key assets in approaching community members and relaying preventative health messages. Our findings reinforce those that demonstrate CHR’s value in promoting access to preventative care, like vaccines, through education and navigation [18,19].

The foremost strength of this study was that 13 CHRs were involved in the development and design of these culturally tailored COVID-19 education materials for their respective communities. Their time and unique perspectives ensured the content and style of the materials resonated with community members. The CHRs were also trained in using REDCap software on iPads as a data collection tool, which was completely new to most of them and hopefully a skill that they can use in the future. The education materials were also viewed by CHR clients as informational, easy to understand, culturally relevant, and were shown to be effective in increasing knowledge about the COVID-19 vaccines.

The limitations of the study are worth noting. First, by the time the study was funded, rates of vaccine uptake were high among tribal partner communities. Despite concerns from historical experiences of misconduct [8] and structural inequalities worsening the impact of COVID-19 on Native nations [8,20], AI/AN led the way in first-dose and full vaccination rates throughout the pandemic. Distinct distribution networks, encouraging and culturally attuned vaccine messaging, and a greater perceived risk are just a couple of explanations for why AI/AN communities may demonstrate such strong vaccine uptake [21]. Although optimistic and persistent, CHRs were challenged to reach people who had not yet been vaccinated. Additionally, at this point in the pandemic, the sense of urgency/interest in COVID-19 was waning, which may have contributed to CHRs’ inability to reach the originally projected sample size for each community. Furthermore, the project required additional time spent in the field on providing education and outreach, potentially interrupting CHRs’ existing tasks. Finally, REDCap was new for many CHRs and some of the NAU staff; time was needed for both partners to work through technical difficulties in using the iPad and REDCap program. Limited understanding of or familiarity with survey instruments (i.e., iPads) resulted in missing data; some respondents unintentionally skipped questions because they did not realize they could scroll through survey questions on the tablets. This resulted in a decreasing *n* over a series of questions during portions of the survey. Future interventions that collect survey data on electronic tablets should take care to explain and demonstrate how participants answer questions to prevent data loss.

An area for future attention is helping community members to understand the benefits of participating in COVID-19 vaccines and therapeutics, as well as other health issues. Many community participants reported a lack of trust in state and federal governments in providing correct health information regarding COVID-19 vaccines. State and federal health bodies, such as the Centers for Disease Control and Prevention and the Arizona Department of Health Services, are free, accurate, and readily available sources of information. Future studies could consider exploring how to rectify the mistrust of these entities in Native communities, and how government bodies can bolster their trustworthiness as well as culturally salient presentations and images regarding health information. We found that other trusted sources of information, such as healthcare providers and CHRs, serve as important resources in educating community members with pertinent and timely health information, especially under changing policy conditions characteristic of the COVID-19 pandemic. Indeed, CHRs have shown great success in indigenous communities when they feel empowered by tribal systems with tools to combat COVID-19, such as testing services, contact tracing, and isolation protocols [22].

## 5. Conclusions

Through the pandemic, society and researchers learned that an effective vaccine does not guarantee uptake. Therefore, advancing vaccine uptake requires the following: (1) acknowledging the existence of systemic racism and inequities across healthcare serving Native nations; (2) engaging local health educators and CHRs in designing acceptable and relevant messaging; (3) improving inadequate healthcare that drives health inequities in Native nations by increasing funding for readily available assets, such as CHR services, in Native communities; and (4) engaging with communities to ensure Native nations reflect, provide feedback, and design solutions to the health needs of their own communities. The results in this study are significant because they demonstrate Native communities’ willingness and openness to vaccinate when provided effective and culturally tailored health information, despite a serious lack of trust in state and federal government bodies. CHRs need to be recognized, funded, and utilized as the assets they are within community healthcare structures. Private–public institutions and future interventions must recognize Native nations’ role in the development and distribution of health information, especially regarding vaccines.

## Figures and Tables

**Table 1 ijerph-20-00031-t001:** Participant demographics at baseline.

	(%) of Total n
Age (years; n = 161)	
18–20	5.0
21–30	18.0
31–40	19.9
41–50	20.5
51–60	16.1
61–70	12.4
71–80	5.6
81–90	2.5
Mean age: 46 y.o.	
Gender (n = 164)	
Man	37.2
Woman	60.4
Nonbinary, genderqueer, or genderfluid	0.6
Prefer not to answer	1.8
Race (n = 165) ^1^	
American Indian or Alaska Native	92.1
White	3.6
Black or African American	0.6
Prefer not to answer	3.0
Other	2.4
Hispanic or Latino (n = 162)	
Yes	10.5
No	82.7
Prefer not to answer	6.8
No. of household members (n = 165)	
1–2	33.3
3–5	50.9
6 or more	15.8
No. of household members who have had COVID-19 (n = 165)	
0	53.3
1–2	27.9
3 or more	18.8
No. of household members vaccinated against COVID-19 (n = 165)	
0	23.6
1–2	48.5
3 or more	27.9

^1^ Respondents allowed to select more than one option; unselected options are omitted.

**Table 2 ijerph-20-00031-t002:** Trusted sources for COVID-19 vaccine information.

Question: How Much Do You Trust Each of These Sources to Provide Correct Information about COVID 19 Vaccines?	(%) of Total n
Your doctor or healthcare provider (n = 162)	
Not at all	7.4
Neutral	29.0
A little	25.9
A great deal	37.7
Your close friends and members of your family (n = 163)	
Not at all	19.6
Neutral	41.7
A little	16.0
A great deal	22.7
People you go to work or class with or other people you know (n = 146)	
Not at all	19.9
Neutral	44.5
A little	17.8
A great deal	17.8
Community Health Worker/Community Health Representative (n = 136)	
Not at all	7.4
Neutral	33.1
A little	23.5
A great deal	36.0
News on the radio, TV, online, or in newspapers (n = 133)	
Not at all	33.8
Neutral	39.1
A little	18.0
A great deal	9.0
Your contacts on social media (n = 127)	
Not at all	48.8
Neutral	29.1
A little	13.4
A great deal	8.7
Your faith leader (n = 128)	
Not at all	20.3
Neutral	37.5
A little	18.0
A great deal	24.2
The U.S. government (n = 127)	
Not at all	34.6
Neutral	52.0
A little	11.0
A great deal	2.4
The Arizona state government (n = 126)	
Not at all	31.7
Neutral	50.0
A little	15.1
A great deal	3.2
Local tribal government (n = 126)	
Not at all	21.4
Neutral	48.4
A little	20.6
A great deal	9.5
Neighboring tribal government (n = 126)	
Not at all	19.8
Neutral	57.1
A little	16.7
A great deal	6.3

**Table 3 ijerph-20-00031-t003:** Vaccine status and confidence.

	(%) of Total n at Baseline	(%) of Total n at Follow-Up
Vaccine Status		
Vaccinated against COVID-19? (n = 165)		
Yes	65.5	67.3
Which vaccine did you get? (n = 109; n = 111) ^1^		
1 dose Moderna or Pfizer vaccine	2.8	16.2
2 doses Moderna or Pfizer vaccine	96.3	79.3
1 dose Johnson & Johnson vaccine	0.0	3.6
Not sure	0.9	0.9
Vaccine Confidence		
Why would you get a COVID-19 vaccine? (n = 165)		
I want to keep my family safe	73.9	50.3
I want to keep my community safe	50.9	29.1
I want to keep myself safe	61.2	47.9
I have an ongoing health problem, like asthma or diabetes	23.6	15.2
My doctor told me to get a COVID-19 vaccine	13.3	6.7
I don’t want to get really sick from COVID-19	31.5	26.1
I want to feel safe around other people	35.8	23.6
I believe life won’t go back to normal until most people get a COVID-19 vaccine	28.5	14.5
Other ^2^	9.1	6.1
Why would you NOT get a COVID-19 vaccine (n = 165)		
I was fully vaccinated against COVID-19 before the education session	x	13.9
I’m allergic to vaccines	2.4	0.6
I have other allergies	6.1	1.8
I don’t like needles	13.3	2.4
I’m not concerned about getting really sick from COVID-19	4.2	0.6
I’m concerned about side effects from the vaccine	38.8	7.9
I don’t think vaccines work very well	4.2	0.6
I don’t trust that the vaccine will be safe	20.6	8.5
I don’t believe the COVID-19 pandemic is as bad as some people say it is	1.8	0.6
I don’t want to pay for it	3.6	0.0
I don’t know enough about how well a COVID-19 vaccine works	34.5	3.6
Other ^3^	15.8	6.7

^1^ (n at baseline; n at follow-up). ^2^ “Other” responses for vaccination included: “Be with family/gatherings; because it’s mandated; want to go back to large gatherings; to go back to work; very sensitive; work mandated. Employment; little to no concern; wanted the $500 incentive.” ^3^ “Other” responses against vaccination included: “Already got it; been vaccinated; beliefs; don’t apply to me; don’t believe; got my shots; health problems; spiritual reasons; haven’t had time. Don’t think the vaccine helps.”.

**Table 4 ijerph-20-00031-t004:** Change in knowledge regarding COVID-19 vaccines between baseline and follow-up.

			**Paired Samples Test**		
			**Paired Differences**		**Significance**
	**Knowledge/Ability to Do the following before and after Intervention**	**Mean**	**Mean**	**Std. Deviation**	**One-Sided *p***
Pair 1 (n = 162)	Explain how the COVID-19 vaccine works ^1^	2.97	−0.383	1.492	<0.001
	Explain how the COVID-19 vaccine works ^2^	3.35			
Pair 2 (n = 163)	Explain the side effects of the COVID-19 vaccine ^1^	3.07	−0.282	1.455	0.007
	Explain the side effects of the COVID-19 vaccine ^2^	3.36			
Pair 3 (n = 137)	Arrange for a COVID-19 vaccine for yourself or family member ^1^	3.62	−0.307	1.292	0.003
	Arrange for a COVID-19 vaccine for yourself or family member ^2^	3.93			

^1^ Baseline assessment. ^2^ 30-day follow-up assessment.

**Table 5 ijerph-20-00031-t005:** Use of and satisfaction with COVID-19 education handouts and information at follow-up.

	(%) of Total n
Have you used the COVID-19 education session handouts or information to talk to a family member or friend about getting the COVID-19 vaccine since our last visit? (n = 163)	
Yes	55.8
No	44.2
Overall, I am satisfied with the COVID-19 vaccine education session. (n = 161)	
Strongly disagree	2.5
Disagree	4.3
Neutral	21.7
Agree	52.2
Strongly agree	19.3
The education session felt relevant to my community and culture. (n = 163)	
Strongly disagree	3.7
Disagree	1.8
Neutral	30.1
Agree	45.4
Strongly agree	19.0
I trust the information provided in the COVID-19 vaccine education session. (n = 152)	
Strongly disagree	2.6
Disagree	5.3
Neutral	25.7
Agree	44.1
Strongly agree	22.4
I trust the educator providing this information. (n = 139)	
Strongly disagree	2.9
Disagree	2.2
Neutral	20.9
Agree	51.1
Strongly agree	23.0
The information shared was easy to understand. (n = 133)	
Strongly disagree	3.0
Disagree	0.0
Neutral	19.5
Agree	57.9
Strongly agree	19.5
The education session gave new information about COVID-19 that I did not previously know. (n = 131)	
Strongly disagree	3.1
Disagree	9.9
Neutral	27.5
Agree	42.7
Strongly agree	16.8

## Data Availability

Data available upon reasonable request to the corresponding author. The data are not publicly available.

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
