# Peer review of "Building Trust and Awareness to Increase AZ Native Nation Participation in COVID-19 Vaccines"

_ijerph, 2022, doi:10.3390/ijerph20010031_

Round 1

Reviewer 1 Report

1. The author should provide more previous studies to support their results in the discussion section.

2. The discussion should suggest the main point of the particular results.  

3. The conclusion section has not shown the main point or the significance of the results. 

Author Response

Thank you for your comments. Please note that we have added five additional resources bolstering the Discussion section (Sabo et al. 2020; Wightman et al. 2022; Sharma, et al. 2019; Yellow Horse et al. 2022, and Foxworth et al. 2021) and elaborated on our main points regarding our results.

The significance of this publication has been made clear in the Conclusion.

Reviewer 2 Report

In Table 3.1. There are results for only being vaccinated or not and the reasons of being vaccinated or not. We can’t see any question about their trial participation decisions. Clinical trial issue had been subjected only in Table 2.

In Table 2, the question “How much do you trust each of these sources to provide correct information about COVID 19 vaccines and clinical trials?” had been asked to participants for vaccines and clinical trials together. Trust for vaccine information would probably differs from trust for vaccine trial. Asking these two components together does not give the correct result. They (vaccines and clinical trials) are needed to be asked separately.

Table 3.2 is also missing vaccine trial issue

Table 1 also has no data for vaccine trial issue

There is “vaccine trials” in the title but missing in results and Tables. “Vaccine hesitancy” is missing in title but results are about this issue dominantly.

As a result; vaccine trials are not subjected in most of the results and subjected in Table 2 with risk of manipulating the result. Vaccine trial is needed to be dropped or revised in the paper. 

Author Response

Thank you for your comments. Please note that any mention of vaccine trial has been dropped from the results. The only mention of vaccine/clinical trials should be where the project team still feels it is applicable to a minor degree, Introduction and briefly in the Methods section. Participants in this study had absolutely no involvement in a clinical trial; a few questions on our survey assessed attitudes and intention towards enrolling in a COVID-19 clinical trial. These questions and corresponding data were ultimately omitted from this publication; this should be better reflected with our revisions.

Reviewer 3 Report

This study presents information on building culturally appropriate prevention strategies including vaccine uptake in American Indian and Alaskan Native communities. This research is important and timely. While much of the COVID-19 information is issued at a large scale level, cultural groups and cultural messaging have largely been ignored. This research fills that gap. 

There are, however, important issues that need to be addressed. I identify two major concerns below. 

First—Are the three groups in this study representative of AI/AN groups more broadly? Does the research apply to other groups outside of AI/AN? How? The groups in this study utilize CHRs. Do other groups, across states for example, also have access to CHRs? It seems the CHRs play a central role in the process, especially in regards to trust. As such, how might the findings be relevant to groups without CHRs? 

In short, what are the implications of the findings beyond the groups in the current research? 

Related, the authors report high rates of vaccination in the sample before the intervention. Is there something about this sample/population that makes them prone to vaccines? A need to protect family/community? I’m wondering if there is a community care perspective in these groups that are not prevalent in the broader population. 

Second — The presentation of data (statistics) raises questions. 

There are different numbers (sample size) across all categories in each of the tables. The authors might consider including only respondents for whom there is full information. Different sample sizes raise important questions, as outlined below. 

Why is there a difference in sample size across pre- and post-survey? 

What are the differences in these samples? 

Might the differences in who is in the sample affect the results? 

i.e. different demographics? Different levels/sources of trust? etc.

It is difficult to make comparisons across groups if the groups being comparing are different

Becomes apples to oranges comparison rather than apples to apples

The results might be different simply because the focus is on two different groups of people 

For example, 65.8% of the sample was vaccinated at the baseline, while 67.3% were vaccinated at the follow-up. Because they are two different samples (272 people, 165 people) it is impossible to gauge whether only 1.5% more people were vaccinated or if the 165 included mostly those who were vaccinated at baseline, or if the 165 includes those who were mostly not vaccinated at baseline. The numbers are difficult to interpret. 

Similar comments across the other tables. In the acceptability and satisfaction table (table 4), the sample size varies across each question. As such, the reader does not know who is in the sample. The first question in table 4 — whether the respondent used information to take to family or friends includes 163 people. This suggest 2 people did not answer any questions (165 people in previous table at time 2.) The last question in table 4 about providing new information, only 131 people answered. What about the other 30 people? Are those 30 people unique in some way that they didn’t answer this question? Is the sample still representative of the demographics? 

Related — demographics are reported for time 1, but not time 2. Did the demographics of the sample change across the two time periods? Trust sources were also asked at time 1, but not time 2. Would the different sample size (and potentially different demographic characteristics of the sample at time 2) reflect different results? How/why? 

Table 4 presents answers regarding the education session; however, only a fraction of the original participants responded. Why? Is there bias in the sample at time 2? 

To reach any sort of conclusion about the research, the data concerns need to be addressed. 

Author Response

Thank you for your comments. Please note that most of your concerns regarding the variations in sample sizes have been addressed upon reanalysis including only those participants who completed both the presurvey and postsurvey. Data from those who completed presurvey only have been omitted. In doing so, the variations between n's is less extreme and is explained in the limitations. Essentially, minor variations in sample sizes (n) are attributed to participants’ oversight of survey questions.

In response to your questions regarding demographics: there were no new participants surveyed at time two and the demographics of the sample did not change between the two time periods. Upon reanalysis, demographics (Table 1) from only those participants who completed both presurvey and postsurvey were included. Additionally, data regarding trusted sources of COVID-19 information (Table 2) were included from only those who completed both presurvey and postsurvey. These questions, however, were asked only at baseline. The questions included in Table 4 were asked only at follow-up; they were not asked in the presurvey as the participants would not have received the education materials until right after.

Finally, the results presented in our manuscript are representative of only the respondents in these three communities, they may not be generalized to other populations. Of the 22 federally recognized Native nations in Arizona, however, 19 operate a CHR program. This has been noted at the beginning of our Methods. We’ve also added more to our Discussion and Conclusion regarding why vaccine uptake may be greater in Native communities and the presence/impact of CHRs in (preventative) healthcare.

We appreciate your time and attention. Thank you.

Round 2

Reviewer 2 Report

Thanks for revisions. But still there are literature review paragraphs on vaccine trial. If you dropped the vaccine trial part totally, you may drop them also ( as lines from 86 to 94)

Author Response

Thank you for your time and attention. Per your recommendations, a paragraph discussing vaccine clinical trial enrollment in the Introduction (lines 77-86), and  corresponding references, has been omitted in efforts to remain concise and relevant.